# Complex Investigation of a Pediatric Haematological Case: Haemophagocytic Syndrome Associated with Visceral Leishmaniasis and Epstein–Barr (EBV) Co-Infection

**DOI:** 10.3390/ijerph15122672

**Published:** 2018-11-27

**Authors:** Giorgia Tascini, Lucia Lanciotti, Lisa Sebastiani, Alessandra Paglino, Susanna Esposito

**Affiliations:** Department of Surgical and Biomedical Sciences, Paediatric Clinic, Università degli Studi di Perugia, 06132 Perugia, Italy; giorgia.tascini@gmail.com (G.T.); lucia.lanciotti@gmail.com (L.L.); sebastianilisa@tiscali.it (L.S.); apaglino@yahoo.it (A.P.)

**Keywords:** Epstein–Barr virus, haemophagocytic syndrome, Leishmania, mononucleosis, visceral leishmaniasis

## Abstract

Background: Visceral leishmaniasis (VL) is an anthropozoonosis caused by an intracellular parasite belonging to the genus *Leishmania*. In the Mediterranean region, *L. donovani* and *L. infantum* are responsible for VL and dogs are the main reservoir. Haemophagocytic lymphohistiocytosis (HLH) represents a complication of VL and consists of unrestrained activation and proliferation of lymphocytes and macrophages, leading to uncontrolled immune activation. Haemophagocytic lymphohistiocytosis may also develop during viral infection, and Epstein–Barr virus (EBV) infection is one of the main HLH causes. Macrophage haemophagocytosis in the bone marrow aspirate is pathognomonic. Case presentation: The case involves a 19-month-old male infant presenting with a high persistent fever with a fluctuating pattern, pancytopaenia, hepatosplenomegaly, and a high triglyceride level. Initial investigations showed an EBV infection. Considering the persistent signs and symptoms, bone marrow aspiration was performed and confirmed the suspicion of HLH. In addition, the presence of *Leishmania* infection was shown. The patient was treated with liposomal amphotericin B and had complete resolution of his symptoms. Conclusion: Diagnosis of VL represents a demanding challenge in endemic and non-endemic areas. Our case demonstrates that leishmaniasis should always be considered in the differential diagnosis in patients presenting with hepatosplenomegaly and cytopaenia with a persistent fever, even in cases of infectious mononucleosis. Moreover, the execution of bone marrow aspiration should not be delayed in order to diagnose and treat at an early stage the potential occurrence of VL, especially if complicated with HLH.

## 1. Introduction

Leishmaniasis is a widely distributed anthropozoonosis; it is endemic to tropical, subtropical, and Mediterranean regions and is caused by an intracellular parasite belonging to the genus *Leishmania*. Phlebotomine sand flies transmit pathogens to humans and animals. According to World Health Organization (WHO) data, the annual incidence is approximately 1.5–2 million new cases per year. In particular, in the Mediterranean region, the annual incidence is 1:239,500/393,600 cases of cutaneous leishmaniasis and 1200–21,000 cases of visceral leishmaniasis [1,2].

The main clinical human leishmaniasis syndromes are visceral, cutaneous, and mucosal leishmaniasis. Approximately 20 pathogenic species infect humans. In particular, *L. donovani* and *L. infantum* are responsible for visceral leishmaniasis (VL) in the Mediterranean region, whereas *L. tropica*, *L. major*, dermotropic *L. infantum* strains, and an *L. infantum*/*L. donovani* hybrid cause cutaneous leishmaniasis [3,4,5] (Table 1). Dogs are the main reservoir. The dissemination of parasites in both the viscera and dermis makes them a perfect source of infection for vectors. In addition, dogs with subclinical disease may maintain the transmission cycle in addition to infections of other wild or domestic animals, such as cats and hares [1,6]. Dog protection with repellents/insecticides (collars or spot and spray formulations) has proven to be the most effective protection strategy.

As a general rule for prophylaxis, the season starts in April and ends in September in the hottest areas of Europe. Vaccination is still an excellent control solution; at present, the combined use of pyrethroids and vaccine is the recommended control system. In Europe, two vaccines are available and can be injected into healthy seronegative dogs six months of age or older [7].

Visceral leishmaniasis is a systemic intracellular infection that is potentially life-threatening and is characterized by a long incubation period (an average of 3–8 months) [8]. The target organs are the liver, spleen, bone marrow, and lymph nodes, which are full of macrophages in which the parasites can proliferate [3]. Typical signs and symptoms of VL are hepatomegaly, splenomegaly, lymphoadenopathies, pancytopaenia, hypoalbuminaemia, hypergammaglobulinaemia, high intermittent fever, anorexia, weight loss, and weakness. The recommended treatment in symptomatic people is the use of liposomal amphotericin B (L-AmB) at a dose of 3 mg/kg/day intravenously (IV) on days 1–5, 14, and 21 (total dose, 21 mg/kg) [9]. Complications of the disease include bacterial infections, secondary haemorrhages, cachexia, multisystem failure, and haemophagocytic lymphohistiocytosis (HLH) [10].

Haemophagocytic lymphohistiocytosis is a disease caused by unrestrained activation and proliferation of lymphocytes and macrophages that leads to uncontrolled immune activation. The pathognomonic characteristic is evidence of macrophage haemophagocytosis in the bone marrow aspirate [11]. Haemophagocytic lymphohistiocytosis comprises the primary form due to hereditary defects and the secondary form (sHLH), which may develop during viral (i.e., Epstein–Barr virus (EBV)), bacterial, protozoal (i.e., *Leishmania*), and fungal infections. Furthermore, malignancies, autoimmune or autoinflammatory diseases, and immunosuppressive therapies can also trigger sHLH. The diagnosis of HLH disease is based on the presence of at least 5 of the 8 clinical, laboratory, and histopathological criteria [12] (Table 2). Treatment of the underlying infection may be sufficient for the resolution of sHLH. In case of treatment failure for the primary infection, HLH-specific therapy (HLH-2004) must be undertaken, which is based on dexametasone, etoposide, and cyclosporine A [12]. Below, we describe a case report for a patient who developed HLH secondary to EBV and VL co-infection.

## 2. Case Presentation

A 19-month-old male infant was admitted to our Paediatric Clinic for dehydration due to rotavirus-associated enteritis based on the results of a stool test. Personal anamnesis showed birth at term, perinatal well-being, and normal psycho-motor development. On day 2 of hospitalization, an intermittent fever appeared associated with a cough and rhinitis. The nasal swab tested positive for adenovirus. We sustained the baby with intravenous fluid and antipyretic therapy. At that point, laboratory tests showed pancytopaenia (white blood cells 4330/mm^3^, 14% neutrophils, haemoglobin 9.3 g/dL, and platelets 70,000/mm^3^) and an increase in the transaminase values. The peripheral blood smear indicated activated lymphocytes and an absence of signs of haemolysis (Coombs test negative, haptoglobin within the normal range). At the resolution of dehydration, enteritis, and fever, we discharged the infant in excellent clinical condition with a diagnosis of pancytopaenia and liver cytolysis during adenovirus and rotavirus co-infection and suggested clinical and biochemical monitoring 7 days later. The baby lived in a small waterfront town by Trasimeno Lake, where his dog was waiting for him.

Three days after returning home, the fever reoccurred without coenaesthesis impairment. A second hospitalization was established at the follow-up visit; the physical examination showed mucocutaneous pallor and hepatosplenomegaly confirmed by ultrasonography in the absence of generalized lymphadenopathies that was associated with an irregular fever for 6 days. The laboratory tests (Figure 1) confirmed the presence of pancytopaenia (white blood cell counts 3370/mm^3^, haemoglobin 7.5 g/dL, and platelets 101,000/mm^3^) and the elevation of inflammatory markers (C reactive protein 6 mg/dL and erythrocyte sedimentation rate 30 mm/1 h). Fibrinogen and the coagulation profile were in range, and the anti-nuclear antibody test was negative. Furthermore, serum immunoglobulin IgA and IgM were normal, but hypergammaglobulinaemia (1855 mg/dL) and high serum ferritin levels (429 ng/mL) were identified. No evidence of echocardiographic abnormalities was found. There was no evidence of malignancy. The peripheral blood immunophenotype, LDH, and uric acid tests were normal, and a chest X-ray was negative for a mediastinal enlargement. An acute EBV infection was found based on positivity for VCA-IgG and VCA-IgM (VCA IgG 409 U/mL, pos >20; VCA IgM 57 U/mL, pos >20; EBNA IgG 3.9 U/mL, pos >5). Epstein–Barr virus DNAemia was not obtained. The remaining serological tests for parvovirus B19, cytomegalovirus, *Bartonella henselae*, HIV, and Widal–Wright were negative, as were the multiple blood cultures and the Mantoux test. At this point, the working diagnosis was mononucleosis complicated by secondary HLH that might be associated with an EBV infection. In fact, the infant presented 4 of 8 diagnostic criteria suggesting HLH: prolonged fever, persistence of hepatosplenomegaly, pancytopaenia (white blood cells 4300/mm^3^, haemoglobin 7.3 g/dL, and platelets 90,000/mm^3^), and elevated triglycerides (546 mg/dL). To obtain a clear diagnosis, bone marrow aspiration was performed. The test revealed the presence of one red blood cell phagocytized by a macrophage but also intracellular and extracellular microorganisms consistent with *Leishmania* amastigotes (Figure 2), which were confirmed by positive serology (indirect immunofluorescence IgG titre of 640, with a cutoff of 1:80). In addition, the blood cell lines were normal except for mild lymphocytic hyperplasia with no evidence of malignancy.

Based on the definitive diagnosis of HLH associated with VL and EBV infection, on day 22 of the fever the child started treatment with L-AmB 3 mg/kg/day for 5 days, followed by two other drug infusions of 3 mg/kg at days 14 and 21 after the beginning of therapy [9]. He showed resolution of fever within 48 h after initiation of treatment, and his clinical condition rapidly improved. The laboratory tests (blood count, triglycerides, and inflammatory indices) gradually normalized. At the follow-up visits, there was no evidence of relapse (Figure 3).

Management of the case was approved by the Ethics Committee of Santa Maria della Misericordia Hospital, Perugia, Italy (2018-PED-01). The patient’s parents provided their written informed consent for the management of their child and the publication of the case report. 

## 3. Discussion

Visceral leishmaniasis is classically considered endemic in the Mediterranean basin, especially in some Italian areas, such as the Tyrrhenian littoral and the southern peninsular regions and islands [3]. Starting in the 1990s, surveillance of dogs and sand flies showed the presence of *L. infantum* even in previously free northern continental areas [13], such as Piedmont [14], Emilia Romagna [15,16], and the foothills of the Apennines zone [17], probably due to changes in the weather and the increasing problem of strays. In the region of Umbria, which is situated in the center of Italy, the exact distribution of the parasite is not known, but the phlebotomines have spread to areas with a high humidity level, such as Trasimeno Lake. In our case, when we collected a more accurate history, we found that the baby lived in a small town near the lake and that the family had a domestic dog, with which the child lived in close contact. However, because his dog was vaccinated for *Leishmania*, the transmission was probably related to the increase in strays.

In the literature, we found three other similar cases in children aged between 9 and 22 months. In particular, in two of these cases, the patients were initially diagnosed with EBV-triggered HLH and were treated with the international treatment protocol (HLH-2004). Due to failure to resolve the clinical picture, further investigations showed *Leishmania* infection. Subsequent therapy with L-AmB led to the resolution of all manifestations of HLH [18,19,20], as described in our case report. Epstein–Barr virus infection is the most common cause of infection-related HLH. A similar case has been described in a 27-year-old male [21]. It is not clear whether the HLH is due to an immunomodulatory effect of VL, allowing the proliferation of EBV-infected lymphocytes, or whether the two events occurred independently from each other. It would be interesting to compare the levels of EBV viremia and antibody titres (especially antibodies against early lytic antigens) in children with HLH alone or HLH + VL to assess whether VL increases EBV viral activity/reactivation. Therefore, in endemic and non-endemic areas, awareness of the possibility of EBV and *Leishmania* co-infection in patients with suspected HLH is important [22]. However, in countries where EBV infection could be perinatal and the humid climate favors transmission of *Leishmania* through mosquito bites, it is urgent to assess the risk of VL complications.

In our case, as in previous reports [18,19,20,21], EBV infection was not treated with any specific drug. Usually, EBV infection is treated in patients with lymphoproliferative disorders and in these cases EBV DNAemia is used to identify those that, due to the absence of effective antivirals, require reduction of immunosuppression and therapy with anti-CD20 monoclonal antibodies [23]. To the best of our knowledge, no experience exists on the use of EBV DNAemia for the identification of patients at risk of EBV-associated HLH as well as on their management on the basis of viral load [11].

## 4. Conclusions

The diagnosis of VL represents a demanding challenge both in endemic and non-endemic areas. Therefore, due to climate change and the problem of strays, *Leishmania* should always be considered in the differential diagnosis in patients presenting with hepatosplenomegaly and cytopaenia with a persistent fever, even in cases of infectious mononucleosis. Moreover, bone marrow aspiration should not be delayed in order to diagnose and treat at an early stage the potential occurrence of VL, especially if complicated with HLH.

## Figures and Tables

**Figure 1 ijerph-15-02672-f001:**
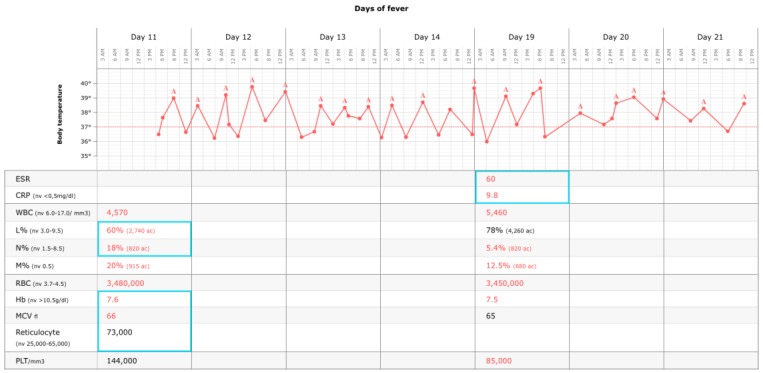
Body temperature and laboratory tests during the second hospitalization (nv: normal value; ac: absolute count; A: acetaminophen). CRP: C reactive protein; ESR: erythrocyte sedimentation rate; Hb: hemoglobin; L%: percentage of lymphocytes; M%: percentage of monocytes; MCV: mean corpuscular volume; N%: percentage of neutrophils; PLT: platelets; RBC: red blood cells; WBC: white blood cells.

**Figure 2 ijerph-15-02672-f002:**
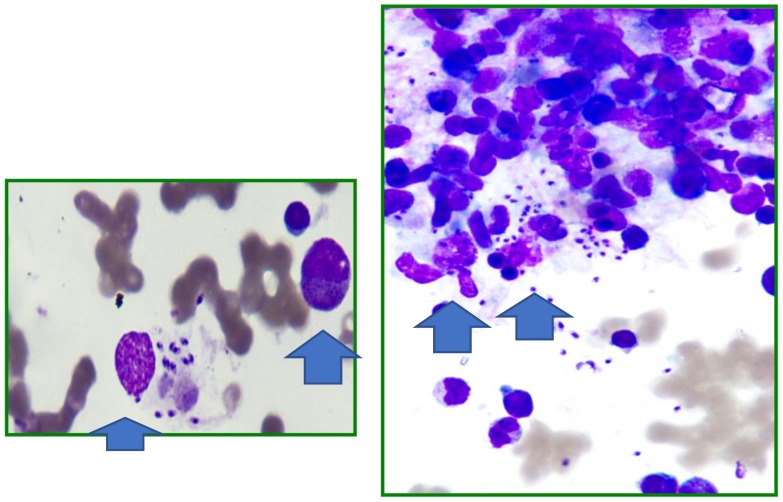
Bone marrow aspiration: red blood cells phagocytized by a macrophage and *Leishmaniae* amastigotes (blue arrows).

**Figure 3 ijerph-15-02672-f003:**
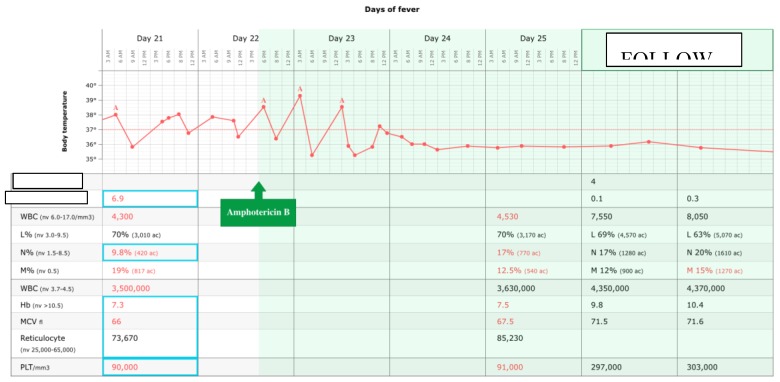
On day 22 of fever, initiation of therapy with Amphotericin B and gradual normalization of laboratory tests.

**Table 1 ijerph-15-02672-t001:** Leishmanial species circulating in the Mediterranean region that are pathogenic for humans.

Complex	Species	Geographical Distribution	Transmission Cycle	Reservoirs
*L. donovani* complex	*L. infantum*	Southern Europe, North Africa, and the Middle East	Zoonotic	Dog ^a^ (*Canis lupus familiaris*), cat ^b^ (*Felis catus*), red fox ^b^ (*Vulpes vulpes*), golden jackal ^b^ (*Canis aureus*), wolf ^b^ (*Canis lupus*), badger ^b^ (*Meles meles*), common genet ^b^ (*Genetta genetta*), pine marten ^b^ (*Martes martes*), wild cat ^b^ (*Felis silvestris*), Iberian lynx ^b^ (*Lynx pardinus*), Iberian hare ^b^ (*Lepus granatensis*), wild rabbit ^b^ (*Oryctolagus cuniculus*), black rat ^b^ (*Rattus rattus*), Norwegian rat ^b^ (*Rattus norvegicus*)
	*L. donovani*	Cyprus and Turkey	Anthroponotic	Human ^c^ (Homo sapiens)

^a^ Proven reservoir: direct or indirect evidence of transmission to the target population (humans); ^b^ A suspected reservoir: evidence of Leishmania spp. infection (culture, serological, or molecular methods) but no evidence of transmission to the target population (humans); ^c^ Anthroponotic cycle has not been confirmed.

**Table 2 ijerph-15-02672-t002:** Haemophagocytic lymphohistiocytosis (HLH) diagnostic criteria.

The HLH Diagnosis Can Be Established if Either 1 or 2 Below is Fulfilled
(1) A molecular diagnosis consistent with HLH
(2) Diagnostic criteria for HLH are fulfilled (5 out of the 8 criteria below)
Fever
Splenomegaly
Cytopaenias (affecting 2 of 3 lineages in the peripheral blood):
Haemoglobin <9 g/dL (in infants < 4 weeks: Haemoglobin <10 g/dL)
Platelets < 100 × 10^9^/L
Neutrophils < 1.0 × 10^9^/L
Hypertriglyceridaemia and/or hypofibrinogenaemia:
Fasting triglycerides ≥265 mg/dL)
Fibrinogen ≤150 mg/dL
Haemophagocytosis in the bone marrow, spleen, or lymph nodes, no evidence of malignancy
Low or absent NK cell activity (according to local laboratory reference)
Ferritin ≥500 µg/L
Soluble CD25 (i.e., soluble IL-2 receptor) ≥2400 U/mL

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
