# Peer review of "Complex Investigation of a Pediatric Haematological Case: Haemophagocytic Syndrome Associated with Visceral Leishmaniasis and Epstein–Barr (EBV) Co-Infection"

_ijerph, 2018, doi:10.3390/ijerph15122672_

Round 1

Reviewer 1 Report

Comment

The authors present a case of HLH associated with EBV infection and visceral Leishmaniasis.

It is a very interesting case with important clinical implications. Namely, the misdiagnosis of a treatable infection and institution of the HLH 2004 protocol could have been deleterious and would not allow the correct treatment of this child.

General comments

In the discussion the EBV infection is not even mentioned. There should be a paragraph addressing this question. Would viral load add something to the diagnosis? Should EBV have been treated? Are antivirals ever indicated?

Specific comments and corrections

Line 27 and 106: delete “the execution of”

Table 1: The “b” and “c” should be upperscript

Table 2: Ferritin is missing the >= symbol

Line 39 and 51: Replace “within range” for “within the normal range”

Line 49: replace mg/dl for g/dL in haemoglobin.

Line 56: replace bulk for enlargement

Figure 2: Identify the elements described with arrows.

Line 95-101: Also mention a similar case described in a 27 year old male:

Gaifer Z, Boulassel MR. Leishmania Infantum and Epstein-Barr Virus Co-Infection in a Patient with Hemophagocytosis. Infect Dis Rep. 2016;8(4):6545. Published 2016 Dec 31. doi:10.4081/idr.2016.6545

Line 100: replace “in non-endemic areas” for “both in endemic and non-endemic areas”

Author Response

The authors present a case of HLH associated with EBV infection and visceral Leishmaniasis.

It is a very interesting case with important clinical implications. Namely, the misdiagnosis of a treatable infection and institution of the HLH 2004 protocol could have been deleterious and would not allow the correct treatment of this child.

Re: Thank you very much for your positive comments. We revised the manuscript according to your recommendations.

General comments

In the discussion the EBV infection is not even mentioned. There should be a paragraph addressing this question. Would viral load add something to the diagnosis? Should EBV have been treated? Are antivirals ever indicated?

Re: We clarified open questions on the role of EBV in HLH as well as on its management (pp. 7, 9, 10).

Specific comments and corrections

Line 27 and 106: delete “the execution of”

Re: Done (pp. 2 and 9).

Table 1: The “b” and “c” should be upperscript

Re: Done (p. 11).

Table 2: Ferritin is missing the >= symbol

Re: Done (p. 12).

Line 39 and 51: Replace “within range” for “within the normal range”

Re: Done (pp. 6 and 7).

Line 49: replace mg/dl for g/dL in haemoglobin.

Re: Done (pp. 6, 7, 12).

Line 56: replace bulk for enlargement

Re: Done (p. 7).

Figure 2: Identify the elements described with arrows.

Re: Done (p. 14).

Line 95-101: Also mention a similar case described in a 27 year old male:

Gaifer Z, Boulassel MR. Leishmania Infantum and Epstein-Barr Virus Co-Infection in a Patient with Hemophagocytosis. Infect Dis Rep. 2016;8(4):6545. Published 2016 Dec 31. doi:10.4081/idr.2016.6545

Re: Done (pp. 9 and 17).

Line 100: replace “in non-endemic areas” for “both in endemic and non-endemic areas”

Re: Done (pp. 9 and 10).

Reviewer 2 Report

The manuscript from Dr Esposito and colleagues reports on a pediatric case of haemophagocytic syndrome associated to Leishmaniasis and EBV co-infection.

EBV infection would be responsible for HLH, a complication of VL.

The manuscript is interesting, as it takes in consideration the pathological combined effect of a co-infection. It is not clear whether the HLH is due to an immunomodulatory effect of VL, allowing proliferation of EBV infected lymphocytes or whether the two events occur independently from each other. Could the authors speculate on the potential mechanism how the two events cooperate? when possible it would be interesting to compare the levels of EBV infection, antibody titre, (specially antibodies against early lytic antigens (EA)) or DNA load, in children with HLH alone or HLH+VL to assess whether VL increases EBV viral activity/reactivation.

The message of the manuscript is relevant as the risk of co-infection could be elevated in other countries then Italy, where EBV infection is perinatal and humid climate favour transmission of Leishmaniasis via mosquitos bites. Therefore, when possible, I would suggest to provide more information about the incidence of  Leishmaniasis and EBV co-infection in different geographical areas to assess the risk of VL complications.    

Author Response

The manuscript from Dr Esposito and colleagues reports on a pediatric case of haemophagocytic syndrome associated to Leishmaniasis and EBV co-infection.

EBV infection would be responsible for HLH, a complication of VL.

Re: Thank you very much for your suggestions. We revised the manuscript accordingly.

The manuscript is interesting, as it takes in consideration the pathological combined effect of a co-infection. It is not clear whether the HLH is due to an immunomodulatory effect of VL, allowing proliferation of EBV infected lymphocytes or whether the two events occur independently from each other. Could the authors speculate on the potential mechanism how the two events cooperate? when possible it would be interesting to compare the levels of EBV infection, antibody titre, (specially antibodies against early lytic antigens (EA)) or DNA load, in children with HLH alone or HLH+VL to assess whether VL increases EBV viral activity/reactivation.

Re: We speculated on the association between VL and EBV infection in HLH (pp. 9 and 10).

The message of the manuscript is relevant as the risk of co-infection could be elevated in other countries then Italy, where EBV infection is perinatal and humid climate favour transmission of Leishmaniasis via mosquitos bites. Therefore, when possible, I would suggest to provide more information about the incidence of  Leishmaniasis and EBV co-infection in different geographical areas to assess the risk of VL complications. 

Re: We highlighted the importance of the assessment of the risk of VL complications in endemic areas (pp. 9 and 10).

Reviewer 3 Report

A well presented case report is presented, with clear and valid observations.

Author Response

A well presented case report is presented, with clear and valid observations.

Re: Thank you very much for your positive evaluation. We revised the manuscript according to other reviewers' suggestions.